# Identification and Pathogenicity of *Fusarium* Fungi Associated with Dry Rot of Potato Tubers

**DOI:** 10.3390/microorganisms12030598

**Published:** 2024-03-16

**Authors:** Olga Gavrilova, Aleksandra Orina, Ilya Trubin, Tatiana Gagkaeva

**Affiliations:** Laboratory of Mycology and Phytopathology, All-Russian Institute of Plant Protection, 196608 St. Petersburg, Russia; orina-alex@yandex.ru (A.O.); ilya.trubin01@mail.ru (I.T.); t.gagkaeva@yahoo.com (T.G.)

**Keywords:** *Fusarium sambucinum*, *F. venenatum*, *F. noneumartii*, *F. mori*, *F. stercicola*, *F. vanettenii*, phylogeny, growth rate, *Solanum tuberosum*, aggressiveness

## Abstract

Dry rot of potato tubers is a harmful disease caused by species of the *Fusarium* genus. Studies on the composition and features of *Fusarium* spp. that cause the disease in Russia are limited. Thirty-one *Fusarium* strains belonging to the *F. sambucinum* species complex (FSAMSC) and *F. solani* species complex (FSSC) were accurately identified using multilocus phylogenetic analysis of the *tef* and *rpb2* loci, and their physiological characteristics were studied in detail. As a result, 21 strains of *F. sambucinum* s. str. and 1 strain of *F. venenatum* within the FSAMSC were identified. Among the analyzed strains within the FSSC, one strain of *F. mori*, four strains of *F. noneumartii*, and two strains of both *F. stercicola* and *F. vanettenii* species were identified. This is the first record of *F. mori* on potato as a novel host plant, and the first detection of *F. noneumartii* and *F. stercicola* species in Russia. The clear optimal temperature for the growth of the strains belonging to FSAMSC was noted to be 25 °C, with a growth rate of 11.6–15.0 mm/day, whereas, for the strains belonging to FSSC, the optimal temperature range was between 25 and 30 °C, with a growth rate of 5.5–14.1 mm/day. The distinctive ability of *F. sambucinum* strains to grow at 5 °C has been demonstrated. All analyzed *Fusarium* strains were pathogenic to potato cv. Gala and caused extensive damage of the tuber tissue at an incubation temperature of 23 °C for one month. Among the fungi belonging to the FSAMSC, the *F. sambucinum* strains were more aggressive and caused 23.9 ± 2.2 mm of necrosis in the tubers on average compared to the *F. venenatum* strain—17.7 ± 1.2 mm. Among the fungi belonging to the FSSC, the *F. noneumartii* strains were the most aggressive and caused 32.2 ± 0.8 mm of necrosis on average. The aggressiveness of the *F. mori*, *F. stercicola*, and especially the *F. vanettenii* strains was significantly lower: the average sizes of damage were 17.5 ± 0.5 mm, 17.2 ± 0.2 mm, and 12.5 ± 1.7 mm, respectively. At an incubation temperature of 5 °C, only the *F. sambucinum* strains caused tuber necroses in the range of 6.7 ± 0.5–15.9 ± 0.8 mm.

## 1. Introduction

A serious problem in potato growing is crop losses due to fungal diseases. Dry rot of potato caused by *Fusarium* fungi is widespread and reduces the food and seed quality of tubers [1,2,3]. *Fusarium* fungi infect tubers in the field and during storage [4]. The vegetative propagation method of potato contributes to the spread of pathogens residing on tubers. Fusarium dry rot on tubers appear as brown spots, sometimes with a dark border, and the internal tissues of the tuber under the influence of fungi often leads to the formation of concentric rings and wrinkling of the periderm [5,6].

According to published information, about 11–13 different species of the *Fusarium* genus are associated with potato dry rot, but the species composition of the fungi depends on environmental conditions [3,7,8]. The *F. sambucinum* species belonging to the *F. sambucinum* species complex (FSAMSC) is often identified in the complex of pathogens of tubers: it dominates among other *Fusarium* fungi in Europe [2,9,10], Asia [1,11,12,13], Africa [3], and North America [14,15,16]. Other predominant species in the mycobiota of potato tubers are members of the *F. solani* species complex (FSSC), *F. oxysporum* species complex, and *F. tricinctum* species complex, as well as fungi from the *F. redolens* and *F. equiseti-incarnatum* species complexes, which were sporadically isolated [2,10,11,12,16,17,18,19].

The aggressiveness of different *Fusarium* spp. to potato plants varies significantly [12,19,20]. The most common pathogenicity assay is the tuber plugging method, which is widely applicable because it allows to obtain unambiguous information on the aggressiveness of strains and reveals the susceptibility of potato cultivars to fungal infection [1,2,3,12,17,21,22,23,24].

New knowledge about *Fusarium* species’ diversity and the effect of environmental conditions on the physiological characters of these fungi is essential to recognize prevention opportunities and develop methods of disease control.

The aim of this study was the molecular identification of *Fusarium* strains isolated from potato tubers with dry rot symptoms and to characterize their physiological characters.

## 2. Materials and Methods

### 2.1. Fusarium Strains

In 2021–2022, tubers with symptoms of dry rot were selected from 46 batches of seed potatoes collected in the Northwest, Central European, South European, Volga, and West Siberian regions of Russia. The isolation of fungi from plant tissues was carried out on a nutrient medium using the conventional mycological method. The tissues were placed on potato–sucrose agar (PSA) containing a mixture of 1 mL/L penicillin–streptomycin solution (HyClone™, GE Healthcare Life Sciences, Wien, Austria) and 0.4 μL/L Triton X-100 (Panreac, Barcelona, Spain). Following incubation at 25 °C, the cultures resembling *Fusarium* fungi were transferred to fresh PSA and synthetic nutrient agar (SNA).

The single-spore *Fusarium* strains (MFG) were deposited in the fungal culture collection of the Laboratory of Mycology and Phytopathology at the All-Russian Institute of Plant Protection (St. Petersburg, Russia). The 31 isolates of *Fusarium* spp. that phenotypically belong to the FSAMSC (22 strains) and the FSSC (9 strains) were selected for a detailed investigation (Table 1). In our study, we also included 12 strains from the collection that were isolated from potato tubers in 2020.

### 2.2. Genomic DNA Isolation, Sequencing, and Phylogenetic Analysis

The *Fusarium* strains were cultured on PSA for 7 days. Genomic DNA was isolated from the fungal mycelium (10–50 mg per strain) using the Genomic DNA Purification Kit (Thermo Fisher Scientific, Vilnius, Lithuania). Amplification of part of the translation elongation factor 1-α gene (*tef*) and RNA polymerase second-largest subunit gene (*rpb2*) was performed as described previously [25,26,27]. Amplicons were sequenced on the ABI Prism 3500 sequencer (Applied Biosystems, Hitachi, Japan) using the BigDye Terminator 3.1 cycle sequencing kit (Applied Biosystems, Foster City, CA, USA).

Consensus sequences of each strain were obtained and manually edited using the Vector NTI Advance 10 program (Thermo Fisher Scientific, Carlsbad, CA, USA). The Basic Local Alignment Search Tool (BLAST) was used to search for similar sequences in the NCBI GenBank database. Sequences mainly representing cultures from the Agricultural Research Service (NRRL, Peoria, IL, USA), the CBS-KNAW culture collection (CBS, Westerdijk Fungal Biodiversity Institute, Utrecht, The Netherlands) and other collections were incorporated into the phylogenetic analysis (Appendix A).

Sequence alignment was performed using MEGA X 10.1 [28]. The FSAMSC and FSSC data sets were analyzed separately. Determination of the best-fitting substitution model and the maximum likelihood (ML) analysis was conducted using IQ-TREE 2 v.2.1.3 program [29]. The TIM2e + I + G4 and TIM2e + R4 models were chosen for the multilocus analysis of the FSAMSC and FSSC data sets, respectively. To infer the phylogenetic relationships among the taxa, maximum parsimony (MP) analysis was conducted using MEGA X 10.1 [28]. Nodal support was assessed by bootstrap analysis based on 1000 replicates. To further infer the phylogenetic relationships among the taxa, Bayesian analysis was conducted with MrBayes 3.2.1 on the Armadillo 1.1 platform [30] using 2,000,000 generations of Markov chain Monte Carlo (MCMC), and the trees were sampled every 1000th generation. The sequence data obtained in this study were deposited in GenBank (Table 1).

### 2.3. Determination of Growth

Twenty of the FSAMSC strains and nine of the FSSC strains were previously grown on PSA for 7 days in the dark at 25 °C. The plates, 85 mm in diameter, with pure PSA were inoculated, and 5 mm disks were taken with a sterile cork-borer from the margin of actively growing fungal culture. The disks were placed, mycelium down, on the surface of the medium in the center of the dish. Each isolate was cultured on PSA at 5, 10, 15, 20, 25, 30, and 35 °C in the dark for 5 days in the Innova 44R thermostat (Eppendorf, Hamburg, Germany).

The size of each colony was measured in two perpendicular directions, and the average diameter of the colony minus the diameter of the inoculation disk was calculated. The growth rate of the strain was calculated as the ratio of the diameter of the fungal colony to the number days of cultivation (mm/day).

Microscopic examination and photography were carried out for colonies grown on SNA with an Olympus BX53 and an Olympus SZX16 microscope (Olympus, Tokyo, Japan).

### 2.4. Pathogenicity Test

The pathogenicity of all strains was assessed by inoculation of the potato tubers under storage conditions at temperatures of 5 °C and 23 °C.

For the experiments, tubers of potato cv. Gala, widely cultivated in Russia, were chosen. Tubers without visible damage and weighing ~40 g were selected, surface-sterilized with 5% sodium hypochlorite for 5–10 min, washed with water, and then air-dried.

Then, the tubers were wounded by a cork borer with a diameter of 5 mm to a depth of 20 mm. The agar disks (4 mm in diameter) were cut out from the fungal cultures grown on PSA for 7 days and placed in the hole, which was subsequently sealed with the excised plug of tuber tissue. At least five tubers were inoculated with one fungal strain, then they were placed in plastic containers (8.5 cm × 19 cm × 31 cm), capped loosely, and incubated at temperatures of 5 °C or 23 °C for 4 weeks. In the control, a disk of pure PSA was used.

After two weeks of incubation, the grown sprouts were removed from the tubers. After 4 weeks, each tuber was cut across the hole, and the width and depth of rotted tissue were measured (mm), calculating the average value. The size of the damage caused by the fungi was assessed for each variant, excluding the average size of the inoculation hole in the control.

### 2.5. Statistical Analysis

The data were analyzed using Microsoft Office Excel 2010 (Microsoft, Redmond, WA, USA) and Statistica 12.0 (StatSoft, Tulsa, OK, USA). The significance of differences among the mean values of groups was estimated by Tukey’s test (95% confidence level).

## 3. Results

### 3.1. Molecular Phylogeny

Multilocus analysis of the *tef* and *rpb2* sequences was used to infer the genetic relationships among *Fusarium* strains. The FSAMSC data set included combined sequences of the 22 analyzed strains as well as 89 reference sequences of *Fusarium* spp. strains and consisted of a total of 1587 characters (666 bp from *tef*, and 921 bp from *rpb2*), among which 963 characters were conserved and 589 were variable (37.1%); 526 characters were parsimony informative (33.1%). *Fusarium nelsonii* strain NRRL 13338 was used as an outgroup (Figure 1). The FSSC data set included combined sequences of 9 analyzed strains and 166 reference sequences of *Fusarium* spp. strains and consisted of a total of 1485 characters (663 bp from *tef*, and 822 bp from *rpb2*), among which 838 characters were conserved and 645 were variable (43.4%); 491 characters were parsimony informative (33.1%). *Geejayessia atrofusca* CBS 125482 was used as an outgroup (Figure 2).

All analyzed strains from the FSAMSC were distributed into the *Sambucinum* clade (Figure 1). Moreover, 21 strains clustered together with the *F. sambucinum* reference strains with high bootstrap support (ML/MP/BP 100/100/1.0). One strain, MFG 70118, formed a clade with the *F. venenatum* reference strains (ML/MP/BP 100/99/1.0).

On the FSSC phylogenetic tree (Figure 2), the four analyzed strains formed a clade with the *F. noneumartii* reference strains (ML/MP/BP 100/73/1.0); two strains, MFG 70108 and MFG 70141, clustered together with the *F. stercicola* reference strains (ML/MP/BP 100/100/1.0); two strains, MFG 70164 and MFG 80216, formed a clade with the *F. vanettenii* reference strains (ML/MP/BP 99/95/0.95); another strain, MFG 70147, formed a clade with the *F. mori* type strain (ML/MP/BP 99/99/1.0).

The appearance of colonies and some micromorphological features of the different *Fusarium* spp. identified in our study are shown in Figure 3. They corresponded to the ranges of morphology characteristics, which are specific to these species [31].

### 3.2. Effect of Temperature on Fungal Growth

At a temperature of 5 °C, only 84% of *F. sambucinum* strains were able to grow on PSA (Figure 4a), and their growth rate averaged 0.9 ± 0.1 mm/day. A single *F. venenatum* strain, like the nine strains of four different species from the FSSC (Figure 4b), did not grow at such low temperature.

In the temperature range of 10–25 °C, the strains of both species from the FSAMSC grew more actively compared to the strains from the FSSC. On the contrary, at 30–35 °C, the diameters of the colonies of the FSSC strains, on average, were significantly larger than those of the colonies of the FSAMSC strains.

The clear optimal temperature for the growth of the FSAMSC strains was noted at 25 °C, when the diameter of the fungal colonies varied in the range of 60.0–79.5 mm. On average, it was 1.2–4.2 times larger than the average size of the colonies determined at a temperature of 10–30 °C. Increasing the temperature to 30 °C led to the slowed down growth of all FSAMSC strains, and at 35 °C, it stopped, with the exception of two strains—*F. sambucinum* MFG 70135 and *F. venenatum* MFG 70118, which continued to grow at rates of 0.8 and 0.9 mm/day, respectively (Table 2).

The optimal growth temperature of the FSSC strains was in the range of 25–30 °C, without significant differences. However, the *F. vanettenii* strains were the slowest-growing compared to the strains of three other species. At the temperature of 35 °C, a significant decrease in the growth rate of all analyzed FSSC strains was detected, up to its complete cessation in 44% of the strains. Both strains of *F. stecicola* included in this study, as well as *F. noneumartii* strain MFG 70176 and *F. vanettenii* strain MFG 80216, were not able grow at the highest temperature in this experiment. The *F. vanettenii* strain MFG 70164 turned out to be relatively tolerant to 35 °C, because its growth rate at that temperature was 5.5 mm/day, which is 1.7–4.2 times higher than the rate of the other growing strains.

### 3.3. Pathogenicity of the Strains

The control tubers remained asymptomatic, and the size of mechanical damage at 23 °C was, on average, 13.5 ± 0.1 mm, and at 5 °C—14.1 ± 0.2 mm.

Upon visual inspection of the tubers inoculated with the FSAMSC, dark depressions of varying sizes were noted around the inoculation hole. Aerial mycelium formulated on the surface of the tuber, and as the affected tissue dried, the surface of the tuber wrinkled in concentric rings. The tuber tissue dried out, and depending on the aggressiveness of the strain, internal cavities of different sizes were formed, separated by a light to dark brown border from the apparently healthy tissue (Figure 5). The surface of the resulting cavity was lined with mycelium and sporulation of the fungus; the color of the aerial mycelium varied from white to salmon-gray. Some FSSC strains, in addition to a distinct lesion cavity, caused extensive maceration of the surrounding tuber tissue, reaching the periderm (Figure 6).

At an incubation temperature of 23 °C, extensive damage to the tuber tissue was observed. The size of tissue necroses caused by the FSAMSC and FSSC strains varied in the ranges of 12.9 ± 1.3–32.5 ± 4.1 mm and 12.5 ± 1.7–33.9 ± 1.0 mm, respectively. The aggressiveness of the strains belonging to two species from the FSAMSC was similar: the *F. sambucinum* strains caused 23.9 ± 2.2 mm of necrosis on average, and *F. venenatum* strain MFG 70118—17.7 ± 1.2 mm (Figure 7).

Among the FSSC strains, the largest tissue necrosis was noted as a result of inoculation of tubers with *F. noneumartii* strains—32.2 ± 0.8 mm on average, compared to the significantly less aggressive strains of *F. mori*, *F. stercicola*, and *F. vanettenii*: the average sizes of the damage caused by these fungi were 17.5 ± 0.5 mm, 17.2 ± 0.2 mm, and 12.5 ± 1.7 mm, respectively.

At an incubation temperature of 5 °C, the sizes of tuber necroses inoculated with *F. sambucinum* strains varied in the range of 6.7 ± 0.5–15.9 ± 0.8 mm, which is, on average, 2.1 times less than that at 23 °C. Only three strains of this species of different geographic origins demonstrated similar aggressiveness at the contrasting temperatures. At 5 °C, the *F. sambucinum* strains turned out to be significantly more aggressive compared to the FSSC strains, which caused necroses that did not exceed 0.3 ± 0.1–3.0 ± 0.7 mm.

## 4. Discussion

To date, information on the species diversity of the pathogens causing Fusarium dry rot of potato tubers in Russia remains scarce. Previously, analysis of the species composition of *Fusarium* fungi isolated from potato tubers with dry rot symptoms revealed four species: *F. sambucinum*, *F. solani*, *F. sporotrichioides*, and *F. oxysporum*, and one more strain could not be identified [32]. Subsequently, the analysis of potato samples collected from storage facilities using the sequencing of *tef* revealed at least 15 different *Fusarium* species, most of which belonged to the *Fusarium oxysporum* species complex [33]. According to our research conducted in 2021–2022, the most common pathogens causing the dry rot of potato tubers in Russia were the species within the FSAMSC [34].

In this study, 31 *Fusarium* strains isolated from potato tubers with symptoms of dry rot, which belong to the FSAMSC and FSSC according to preliminary morphological analysis, were randomly chosen from the fungal culture collection. Multilocus analysis of the combined *tef* and *rpb2* sequences was used to infer the taxonomic status of these strains. The topologies of the constructed trees for the FSAMSC and FSSC by different methods are concordant with the phylogenetic relationships among *Fusarium* species reconstructed previously [35,36,37,38,39,40,41,42]. The phylogenetic analysis allowed for accurately identifying the *Fusarium* species: among the analyzed FSAMSC strains, most of them belong to *F. sambucinum* and one to *F. venenatum*, and among the FSSC strains, four species (*F. mori*, *F. noneumartii*, *F. stercicola*, and *F. vanettenii*) were detected.

*Fusarium sambucinum* was one of the first described species in the *Fusarium* genus. Later, the significant intraspecific diversity among *F. sambucinum* strains was revealed, and three morphologically similar species—*F. sambucinum* s. str., *F. venenatum*, and *F. torulosum*—were described [31,43,44]. Currently, the *Sambucinum* clade within the FSAMSC includes 6 described species and at least 10 phylogenetic lineages [40].

Among the analyzed fungi within the FSAMSC, 21 strains of *F. sambucinum* s. str. and one strain of *F. venenatum* were identified. The last species was previously isolated from potato tubers in Algeria [3] and Poland [2] and was also found in the soil and mycobiota of various cereals [40]. In Russia, this is the first finding of *F. venenatum* in the mycobiota of potato; it was previously identified only in oat and wheat grain [45].

*Fusarium solani* was first isolated from potato with dry rot in 1842 in Germany and described as the pathogen causing this disease. Currently, there are more than 70 recognized phylogenetic species of the FSSC, while some of them do not yet have formal descriptions or Latin binomial names [46,47,48,49]. The *Fusarium solani* species complex is the most controversial group of fungi, both in terms of its intraspecific diversity and its degree of relatedness to the genus *Fusarium* s. str. [39,47,50,51]. These fungi are cosmopolitan and not confined to one species or family of host plants [38,52].

In this study, among the strains distinguished within the FSSC, one strain of *F. mori*, four strains of *F. noneumartii*, and two strains of both *F. stercicola* and *F. vanettenii* species were identified. This is the first record of *F. mori* on potato as a novel host plant and the first finding of this species in Russia. In previous studies, *F. mori* was isolated from mulberry in Japan [52] and China [53] and was also found in the mycobiota of tomato stems in Japan [54]. *Fusarium noneumartii* and *F. stercicola* were also identified in Russia for the first time. Previously, *F. noneumartii* was isolated from potatoes in Israel and from tomatoes in the USA [52], as well as from the roots of orange trees in South Africa [55]. *F. stercicola* has been isolated from plant debris in Switzerland [56], from soil in Europe [38,57] and in China [41], and also has been found on *Solanum tuberosum* and other host plants in New Zealand [58] and in nematodes [59]. *F. vanettenii* was previously isolated from potato tubers in Russia [33,60], as well as from tomato [61], legumes [42], and soil [38]. The existing information indicates a potentially wide distribution of *F. mori*, *F. noneumartii*, *F. stercicola*, and *F. vanettenii* and other FSSC species in the mycobiota of potato and widely cultivated crops.

Temperature significantly affects the adaptability of fungi in nature. Our results demonstrate that a temperature of 5 °C was favorable for the growth of only the *F. sambucinum* strains, whereas all the strains of the other analyzed five species were not able grow at this low temperature. These results are in agreement with previously obtained data in Tunisia showing the ability of *F. sambucinum* to grow at temperatures below 5 °C, contrary to *F. solani* [62]. This tolerance to extremely low temperatures can give *F. sambucinum* an advantage over other representatives of tuber mycobiota under storage conditions, where the temperature does not exceed 4–10 °C.

In our study, 25 °C was the optimal temperature for growth of the *F. sambucinum* and *F. venenatum* strains, and 25–30 °C for the growth of FSSC strains. Thus, this confirms the previous data showing that at temperatures below 25 °C, *F. sambucinum* grows faster than *F. solani*, but high temperatures (30–35 °C) are more favorable for the growth of the latter [63]. Increasing the temperature to 35 °C led to a stop in the growth of 95% of the *F. sambucinum* strains; only one strain, MFG 70135, from Stavropol Krai, which is the most southern region in this study, was able to grow at a rate of 0.8 mm/day. The cultivation of the FSSC strains at 35 °C can be used as an additional marker for the separation of *F. stercicola* from phylogenetically related species of the FSSC; both strains of this species completely stopped their growth at this high temperature. Inconsistency of the *F. vanettenii* strains in response to 35 °C was established; one strain, MFG 70164, turned out to be relatively tolerant, but the other strain, MFG 80216, did not grow.

All analyzed strains caused significant damages of the tubers at 23 °C, but the largest tissue necroses were induced by the *F. noneumartii* and *F. sambucinum* strains. The *F. venenatum*, *F. mori*, *F. stercicola*, and especially the *F. vanetenii* strains can be characterized as less aggressive. Previously, the higher aggressiveness of the *F. sambucinum* strains to potato tubers, compared to the other fungi, including members of the FSSC, has been repeatedly noted [1,2,3,12,19,23,62,63].

The characterization of the aggressiveness of the FSSC isolates (*F. solani* f. sp. *eumartii*, *F. coeruleum*, and *F. eumartii*) to potato revealed their differences: some isolates caused sunken dry lesions and vascular discoloration on tubers, while others did not cause lesions on any of the inoculated tubers and showed minimal growth on plant tissue [64]. In that study, *F. sambucinum* isolate caused sunken lesions on the tubers. Our observations showed, under inoculation with FSSC fungi, softening and moistening of the internal tissue of tubers sometimes occurred, which is not very similar to the classic symptom of Fusarium dry rot and should be taken into account when visually diagnosing potato diseases.

The aggressiveness of the strains belonging to four species of the FSSC varied significantly at 23 °C. Previously, the significant differences in the aggressiveness among *F. solani* s. lat. strains to potato tubers (20 °C, 3 weeks) was revealed [21,24]. According to [65], the pathogenicity of *F. solani* strains varies from highly aggressive to completely non-pathogenic. However, the lack of molecular identification of the strains in those studies does not allow us to conclude on the interspecific differences in *F. solani* s. lat. However, in our study, the significantly higher aggressiveness of the *F. noneumartii* strains compared to the *F. mori*, *F. stercicola*, and *F. vanetenii* strains was revealed.

At 5 °C, the *F. sambucinum* strains turned out to be significantly more aggressive compared to the other analyzed fungi, which is consistent with earlier results [20,62]. Thus, the *F. sambucinum* strains turned out to be consistently aggressive and capable of causing dry rot in tubers at a wide range of temperatures close to growing and storage conditions. The inability of the FSSC strains to grow at 5 °C indicates the lack of infection development in the inoculated tubers. This highlights the key effect of environmental conditions on the physiological characters of pathogens and their interaction with host plants.

The pathogenicity of fungi is a complex process that depends on the expression of many genes and plant resistance mechanisms [5], e.g., the cutinase activity in the *F. solani* strains correlated with their pathogenicity to potato tubers [66]. *Fusarium* species associated with potato dry rot are known to produce different metabolites, some of which are phytotoxins [67,68]. Previously, in the extract obtained from potato tubers infected with the *F. sambucinum* strain, 18 compounds were found, including 5 mycotoxins/phytotoxins (aurofusarin, beauvericin, sambutoxin, emodin, and fusaric acid) and 2 specific fungal compounds (bikaverin and deoxygerfelin), whereas after *F. sambucinum* cultivation on malt extract agar, only 8 compounds, including 2 mycotoxins (beauvericin and diacetoxyscirpenol (DAS)), were detected [69]. The accumulation of DAS in the tubers inoculated with *F. sambucinum* was analyzed under different storage conditions, and an increase temperature from 10 °C to 20 °C over a period of 30 and 50 days led to an increase in the DAS amounts found in the tubers [70]. Enniatins are mycotoxins produced by different *Fusarium* fungi, including those that are associated with potato diseases, causing necrosis of the potato tissue *in vitro*, but these mycotoxins are not essential for the successful infection of tuber tissue by fungal strains [5]. Loss of the ability to produce enniatins in the strain was correlated with it causing less tuber necrosis. Conversely, the *ESYN1* gene overexpression led to an increase in enniatin production, as well as more tuber necrosis [71].

Thus, with a large collection of *Fusarium* strains isolated from potato tubers obtained from different regions, it would be useful to establish the proportions of fungal species in the distant regions with different environmental conditions. In the future, it would also be interesting to reveal the specialization of the identified *Fusarium* species not only to different potato varieties but also to other crops, as well as to determine the profiles of metabolites and their role in the interaction with the host plant.

## 5. Conclusions

*Fusarium* strains isolated from potato tubers with dry rot symptoms collected from different regions of Russia were accurately identified using multilocus phylogenetic analysis of the *tef* and *rpb2* loci. The *Fusarium sambucinum* and *F. venenatum* species were identified among the FSAMSC strains, and the *F. mori*, *F. noneumartii*, *F. stercicola*, and *F. vanettenii* species were present among the FSSC strains. This is the first finding of *F. mori* on potato as a novel host plant, and the first detection of the *F. noneumartii* and *F. stercicola* species in Russia. The clear optimal temperature for the growth of the FSAMSC strains was determined at 25 °C, whereas for the FSSC strains, the optimum was in the range of 25–30 °C. The ability of the *F. sambucinum* strains to grow at 5 °C distinguished them from the other analyzed fungi. All analyzed *Fusarium* strains were pathogenic to potato cv. Gala and caused extensive lesions in the tuber tissue at an incubation temperature of 23 °C. At an incubation temperature of 5 °C, only the *F. sambucinum* strains caused necroses, whereas the other *Fusarium* strains were not able to infect the tuber tissue. Low-temperature storage can slow down the process of infection by *Fusarium* species, but violation of storage conditions can lead to significant potato losses. *Fusarium* species producing a wide spectrum of metabolites makes it is necessary to pay special attention to the contamination of potato seed tubers with these fungi, since, even if storage conditions are met, the presence of infection poses a serious threat to future yield.

## Figures and Tables

**Figure 1 microorganisms-12-00598-f001:**
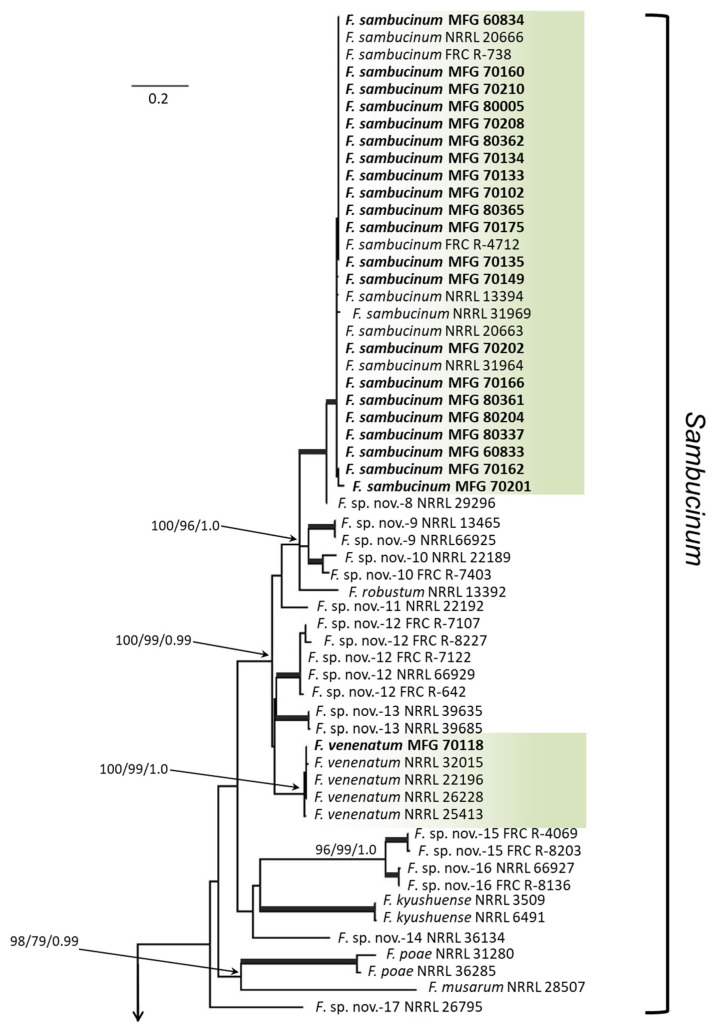
Maximum likelihood (ML) phylogenetic tree based on DNA sequence data from *tef* and *rpb2* loci of *Fusarium* spp. from *Fusarium sambucinum* species complex. ML and maximum parsimony (MP) bootstrap support values > 70%, followed by Bayesian posterior probability (BP) scores > 0.95 are shown at the nodes. Thickened lines indicate ML/MP of 100 and a BP of 1.0. The studied strains are in bold. The tree was rooted on sequences of *F. nelsonii* strain NRRL 13338.

**Figure 2 microorganisms-12-00598-f002:**
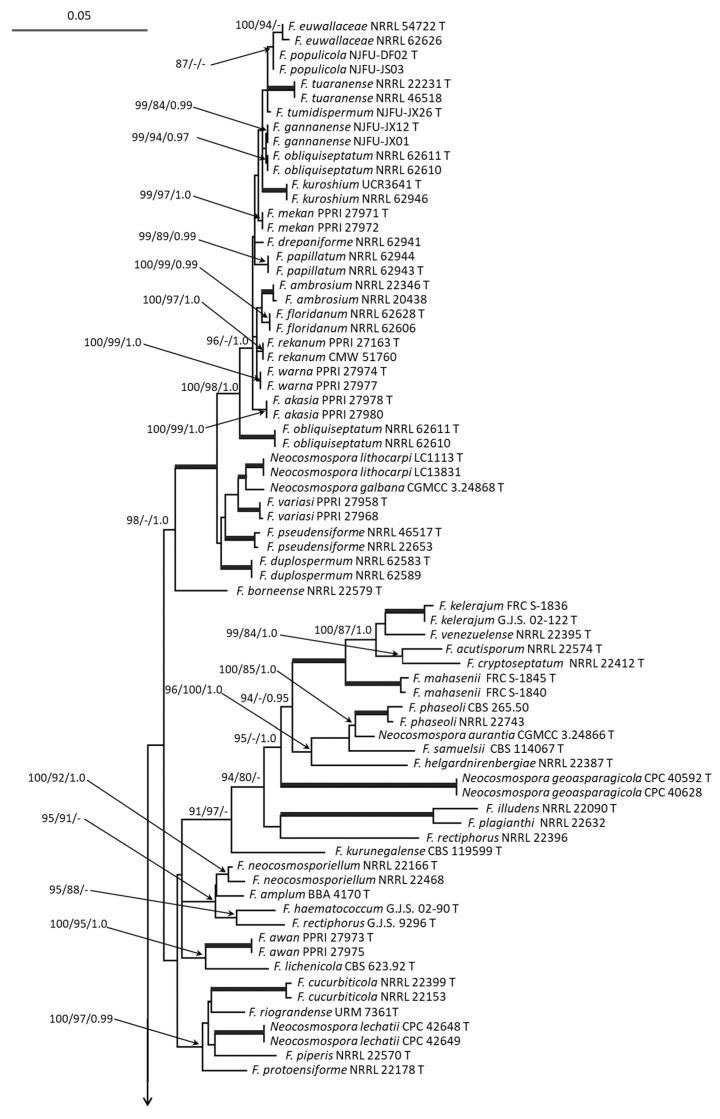
Maximum likelihood (ML) phylogenetic tree based on DNA sequence data from *tef* and *rpb2* loci of *Fusarium* spp. from *Fusarium solani* species complex. ML and maximum parsimony (MP) bootstrap support values > 70%, followed by Bayesian posterior probability (BP) scores > 0.95 are shown at the nodes. Thickened lines indicate ML/MP of 100 and BP of 1.0. The studied strains are in bold. The tree was rooted on sequences of *Geejayessia atrofusca* strain CBS 125482.

**Figure 3 microorganisms-12-00598-f003:**
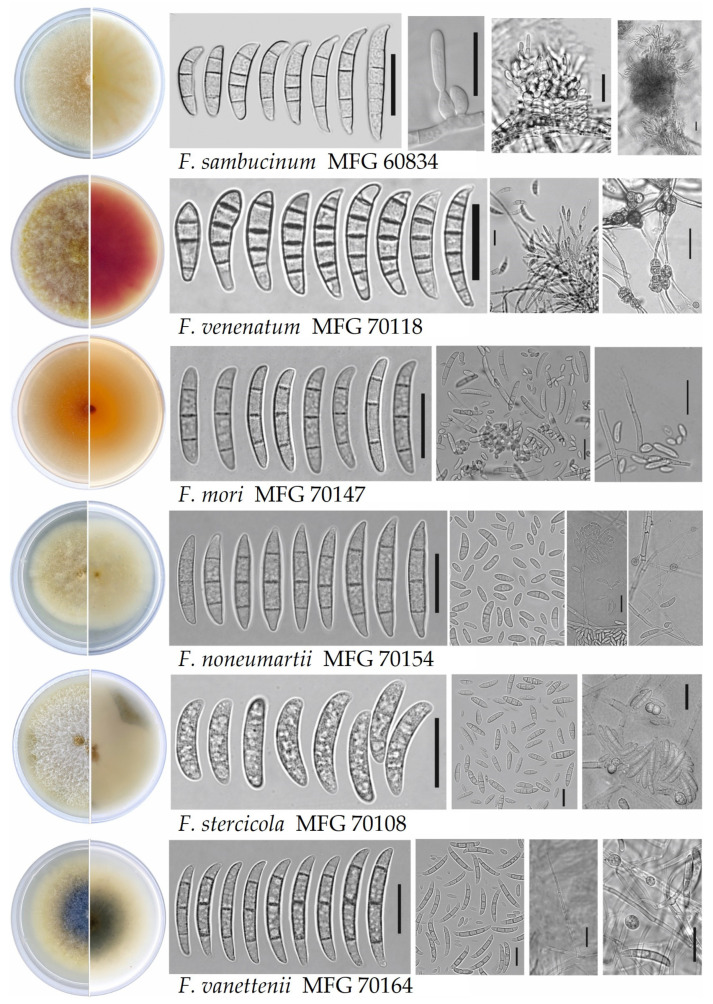
Colony morphology on potato sucrose agar and the micromorphological characteristic on synthetic nutrient agar of analyzed *Fusarium* fungi isolated from potato tubers (25 °C in the darkness). Scale bars = 20 μm.

**Figure 4 microorganisms-12-00598-f004:**
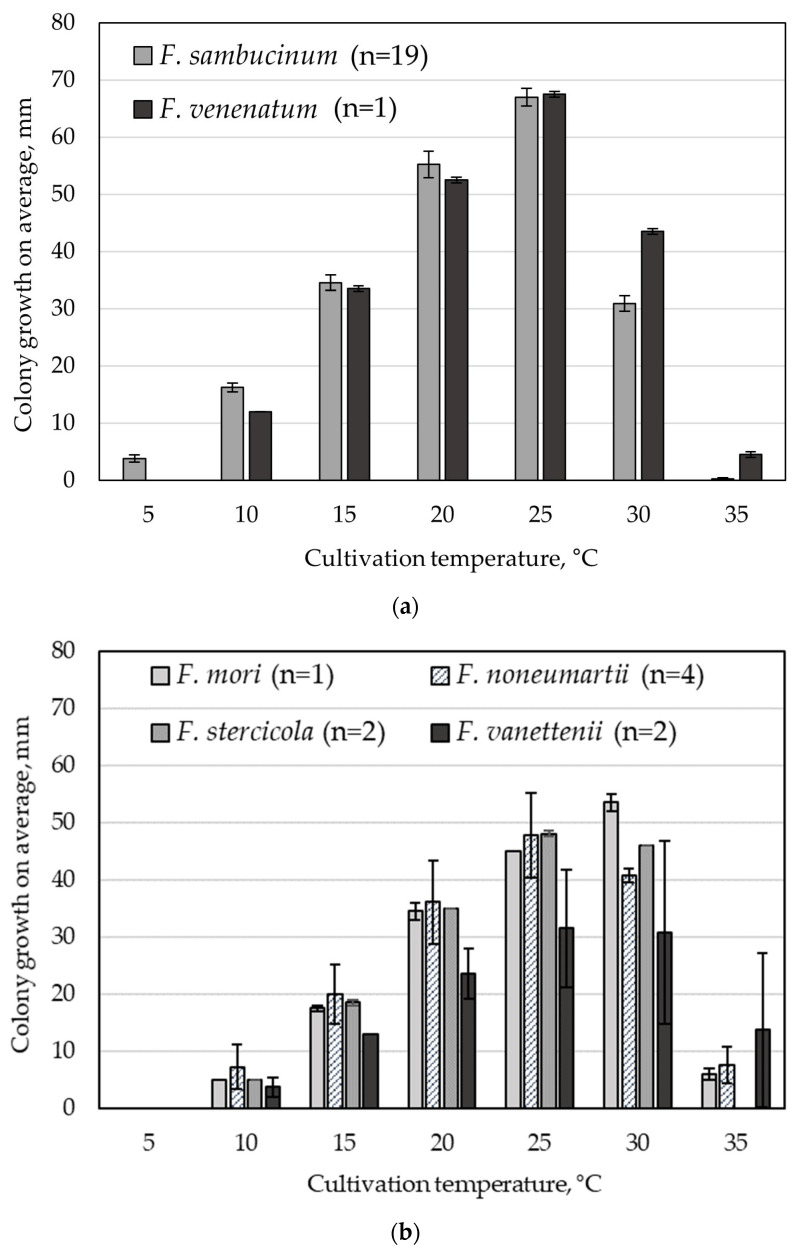
Effect of temperature on the growth of the strains identified within the *Fusarium sambucinum* species complex (**a**) and *Fusarium solani* species complex (**b**) (potato–sucrose agar, 5 days, in the darkness). The bars indicate the average values, and the intervals are the confidence interval at a significance level of *p* ˂ 0.05.

**Figure 5 microorganisms-12-00598-f005:**
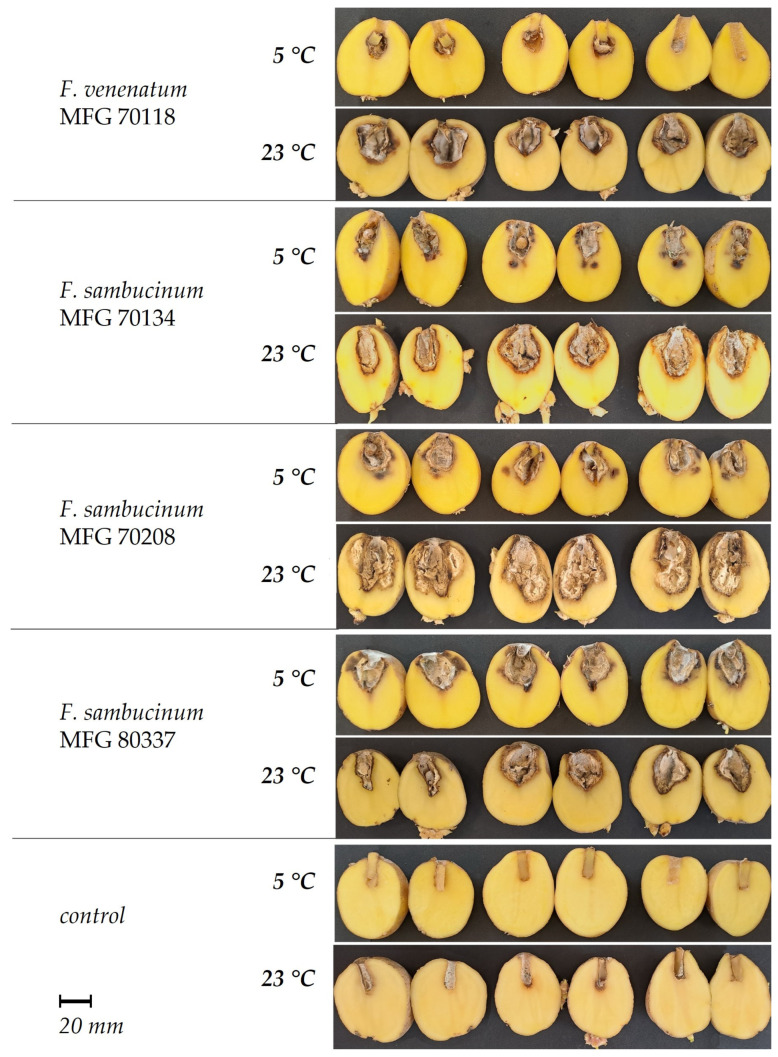
Pathogenicity of the strains within *Fusarium sambucinum* species complex in tubers of potato cv. Gala at two incubation temperatures (4 weeks).

**Figure 6 microorganisms-12-00598-f006:**
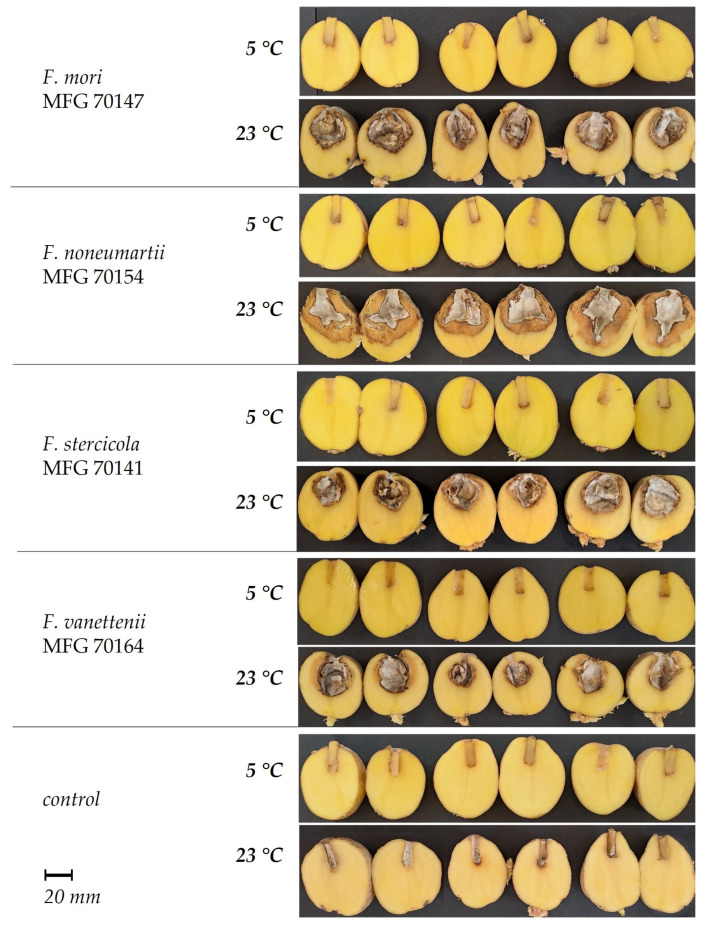
Pathogenicity of the strains within *Fusarium solani* species complex in tubers of potato cv. Gala at two incubation temperatures (4 weeks).

**Figure 7 microorganisms-12-00598-f007:**
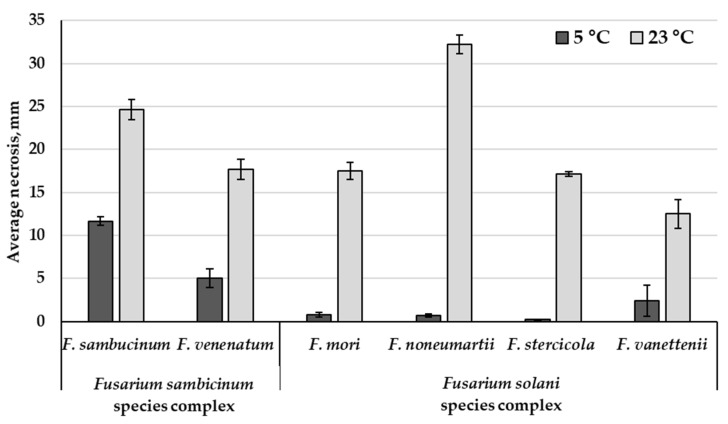
Pathogenicity of *Fusarium* fungi in tubers at two incubation temperatures (potato cv. Gala, 4 weeks, in the darkness). The bars indicate the average values, and the intervals are the confidence interval at a significance level of *p* ˂ 0.05.

**Table 1 microorganisms-12-00598-t001:** *Fusarium* spp. included in the study.

Species	Species Complex	Strain ID	Origin	Year	GenBank Accessions	
					*tef*	*rpb2*
*F. sambucinum*	FSAMSC *	MFG 60833	Russia: Vologda region	2020	OR020701	OR727754
*F. sambucinum*	FSAMSC	MFG 60834	Russia: Vologda region	2020	OR020702	OR727755
*F. sambucinum*	FSAMSC	MFG 70102	Russia: Novgorod region	2020	OR020704	OR727756
*F. sambucinum*	FSAMSC	MFG 70133	Russia: Yaroslavl region	2020	OR020710	OR727758
*F. sambucinum*	FSAMSC	MFG 70134	Russia: Yaroslavl region	2020	OR020711	OR727759
*F. sambucinum*	FSAMSC	MFG 70135	Russia: Stavropol region	2020	OR020712	OR727760
*F. sambucinum*	FSAMSC	MFG 70149	Russia: Samara region	2020	OR020717	OR727761
*F. sambucinum*	FSAMSC	MFG 70160	Russia: Moscow region	2021	OR020724	OR727762
*F. sambucinum*	FSAMSC	MFG 70162	Russia: Chuvashia	2021	OR020725	OR727763
*F. sambucinum*	FSAMSC	MFG 70166	Russia: Chuvashia	2021	OR020728	OR727764
*F. sambucinum*	FSAMSC	MFG 70175	Russia: Kaluga region	2021	OR020730	OR727753
*F. sambucinum*	FSAMSC	MFG 70201	Russia: Vologda region	2021	OR020734	OR727765
*F. sambucinum*	FSAMSC	MFG 70202	Russia: Udmurtia	2021	OR020735	OR727766
*F. sambucinum*	FSAMSC	MFG 70208	Russia: Ryazan region	2022	OR020736	OR727767
*F. sambucinum*	FSAMSC	MFG 70210	Russia: Ryazan region	2022	OR020737	OR727768
*F. sambucinum*	FSAMSC	MFG 80005	Russia: Novgorod region	2020	OR020738	OR727769
*F. sambucinum*	FSAMSC	MFG 80204	Russia: Moscow region	2021	OR020739	OR727770
*F. sambucinum*	FSAMSC	MFG 80337	Russia: Tula region	2022	OR020741	OR727771
*F. sambucinum*	FSAMSC	MFG 80361	Russia: Omsk region	2022	OR020742	OR727772
*F. sambucinum*	FSAMSC	MFG 80362	Russia: Omsk region	2022	OR020743	OR727773
*F. sambucinum*	FSAMSC	MFG 80365	Russia: Omsk region	2022	OR020744	OR727774
*F. venenatum*	FSAMSC	MFG 70118	Russia: Leningrad region	2020	OR020706	OR727757
*F. mori*	FSSC **	MFG 70147	Russia: Samara region	2020	OR020716	OR727777
*F. noneumartii*	FSSC	MFG 70154	Russia: Bashkiria	2021	OR020719	OR727778
*F. noneumartii*	FSSC	MFG 70155	Russia: Bashkiria	2021	OR020720	OR727779
*F. noneumartii*	FSSC	MFG 70176	Russia: Moscow region	2021	OR020731	OR727781
*F. noneumartii*	FSSC	MFG 70177	Russia: Moscow region	2021	OR020732	OR727782
*F. stercicola*	FSSC	MFG 70108	Russia: Pskov region	2020	OR020705	OR727775
*F. stercicola*	FSSC	MFG 70141	Russia: Stavropol region	2020	OR020715	OR727776
*F. vanettenii*	FSSC	MFG 70164	Russia: Chuvashia	2021	OR020726	OR727780
*F. vanettenii*	FSSC	MFG 80216	Russia: Pskov region	2021	OR020740	OR727783

* FSAMSC—*Fusarium sambucinum* species complex; ** FSSC—*Fusarium solani* species complex.

**Table 2 microorganisms-12-00598-t002:** Growth rates of strains within the *Fusarium sambucinum* species complex and *Fusarium solani* species complex (mm/day) at different temperatures (PSA, in the darkness).

SC *	*Fusarium* spp. (No. of Strains)	Temperature, °C						
		5	10	15	20	25	30	35
FSAMSC	*F. sambucinum*	0.8 ± 0.1 **	3.2 ± 0.2	6.9 ± 0.3	11.0 ± 0.5	13.4 ± 0.3	6.2 ± 0.3	0
	(*n* = 19)	(0–1.8)	(2.0–4.4)	(5.1–8.8)	(8.0–13.8)	(11.6–15.0)	(3.6–8.3)	(0–0.8)
	*F. venenatum*	0	2.4	6.7 ± 0.1	10.5 ± 0.1	13.5 ± 0.1	8.7 ± 0.1	0.9 ± 0.1
	(*n* = 1)		(2.4; 2.4)	(6.6; 6.8)	(10.4; 10.6)	(13.4; 13.6)	(8.6; 8.8)	(0.8; 1.0)
FSSC	*F. noneumartii*	0	1.5 ± 0.8	4.0 ± 1.0	7.2 ± 1.5	9.6 ± 1.5	8.2 ± 0.2	1.5 ± 0.6
	(*n* = 4)		(0.6–3.8)	(3.0–7.2)	(5.5–11.7)	(8.0–14.1)	(7.5–8.6)	(0–3.2)
	*F. stercicola*	0	1.0	3.7 ± 0.1	7.0	9.6 ± 0.1	9.2	0
	(*n* = 2)		(1.0; 1.0)	(3.6; 3.8)	(7.0; 7.0)	(9.5; 9.7)	(9.2; 9.2)	
	*F. vanettenii*	0	0.8 ± 0.3	2.6	4.7 ± 0.9	6.3 ± 2.1	6.2 ± 3.2	2.8 ± 2.7
	(*n* = 2)		(0.4; 1.1)	(2.6; 2.6)	(3.8; 5.6)	(4.2; 8.4)	(2.9; 9.4)	(0; 5.5)
	*F. mori*	0	1.0	3.5 ± 0.1	6.9 ± 0.3	9.0	10.7 ± 0.3	1.2 ± 0.2
	(*n* = 1)		(1.0; 1.0)	(3.4; 3.6)	(6.6; 7.2)	(9.0; 9.0)	(10.4; 11.0)	(1.0; 1.4)

* SC—species complex: FSAMSC—*Fusarium sambucinum* species complex; FSSC—*Fusarium solani* species complex. ** The average value with a confidence interval at a significance level of *p* ˂ 0.05; the range of values is indicated in parentheses.

## Data Availability

The data are contained within the article.

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
