# Peer review of "Identification and Pathogenicity of *Fusarium* Fungi Associated with Dry Rot of Potato Tubers"

_microorganisms, 2024, doi:10.3390/microorganisms12030598_

Round 1
Reviewer 1 Report
Comments and Suggestions for Authors
line 9-10: the word "that" should be deleted florm line 9, and used in line 10
line 37: ...propagation method
line 42 and 43: I think it could be re-written
line 50: re-write the sentence, in line 51: "method"
line 55, 56: re-write the sentence
line 65: replace "common"
line 105-106: revise the sentence
growth rate determination from my point of view is not well explained
line 114- --- and then their.. please revise that
line 212: its growth rate at... that temperature.. was...
line 289: I think that this sentence could be improved
line 311: please re-write this sentence
line 315: revise the sentence
line 317: during storage contditions?
line 332: following.. by??
line 338, remove "of", and in line 339, replace "and" for "while"
line 378: ... an increase of...
line 380, remove is
line 392: the sentence is not clear
In general I think the manuscript is very interesting and well organized. however, I sugget the authors to read it carefully in order to correct some little spealling mistakes.
I think that tables, figures are very usefull
Comments on the Quality of English LanguageAll comments are in the previuos box.
Author Response
Dear Reviewer,
Thank you very much for your valuable help in improving of our manuscript. Based on your comments, we checked the manuscript thoroughly and made relevant changes.
Please see the attachment.

Reviewer 2 Report
Comments and Suggestions for Authors
This manuscript was well written with an excellent standard of English. The methodology and analyses were performed appropriately, and in line with current global standards. The ability of F. sambucinum strains to cause nectric growth at 5C was an interesting finding.
Author Response
Dear Reviewer,
Thank you for your time in reviewing our manuscript and the high appreciation of our work!
Reviewer 3 Report
Comments and Suggestions for Authors
I found this to be a thorough investigation with robust data, although I have a few general concerns that need to be addressed before the manuscript can be accepted.
Without completing Koch’s Postulates (i.e., reisolation from inoculated tuber tissues), the authors can not make the claim of the new host relationship for F. mori, F. noneumartii, and F. stercicola. If the fungi were in fact reisolated successfully, this needs to be stated in the manuscript. If they were not, no claims about a first record can be made.
The English writing of the manuscript (word choice, grammar, sentence structure) needs to be improved considerably before the manuscript can be considered for publication in an international journal. The authors should be encouraged to consult with one of the commercial English language editing services to improve their manuscript prior to resubmission.
Typically I would be very concerned about conclusions drawn from experiments that were not repeated independently (i.e., not confirmed in two separate experiments), such as was the case with the aggressiveness assay on potato tubers. However, I believe that the large number of isolates included in these assays provided robust results and therefore compensate for the lack of separate experiments.
Additional comments and suggestions:
31: Delete “temperature” and “necrosis” from the list of keywords. Add “Solanum tuberosum”.
33-60: The Introduction is too cryptic and does not include sufficient information to set the stage for this investigation. For example, how does Fusarium dry rot rank compared with other tuber diseases of potato in Russia? What are typical percent losses associated with the disease? How is the disease managed (in general terms)? What was known about the dry rot-causing Fusarium species composition and their aggressiveness prior to the initiation of the current study?
63: This section says that tubers were collected in 2021 and 2022, but Table 1 includes several strains from 2020.
64: Russia is a vast territory. From which regions specifically were these potato tuber batches obtained?
66: What antibiotic(s) specifically were included in the PSA medium?
103: All 31 strains?
114: Again all 31 strains?
124: A “cuvette” is a very small vial used to take measurements with a spectrophotometer for example. Some other container must have been used to store intact potato tubers.
383: At the end of the Discussion, add a brief paragraph outlining the next steps needed in this research.
409: Only summarized data, but not raw data, is contained in the article. Add a statement of how the interested reader can get access to the raw data.
Table 1: FSAMSC and FSSC need to be explained in a footnote so that the table can stand on its own.
Figure 1: FSAMSC, ML, MP, and BP need to be explained in the figure caption so that the figure can stand on its own.
Figure 2: FSSC, ML, MP, and BP need to be explained in the figure caption so that the figure can stand on its own.
Figure 3: Spell out PSA and SNA in the figure caption (I do not think SNA was mentioned in Materials & Methods). Explain the scale bars in the micrographs.
Figure 4: Again, spell out all the acronyms. Also mention how many isolates of each species were used to calculate the means. This could be done by adding the isolate numbers in parentheses after each species name in the figure (similar to how it is presented in Table 2). Also indicate whether the error bars are SD or SE.
Table 2: Explain the acronyms in a footnote.
Figures 5 and 6: Spell out all the acronyms.
Figure 7: Spell out all the acronyms. Also indicate whether the error bars are SD or SE.
Comments on the Quality of English LanguageThe English writing of the manuscript (word choice, grammar, sentence structure) needs to be improved considerably before the manuscript can be considered for publication in an international journal. The authors should be encouraged to consult with one of the commercial English language editing services to improve their manuscript prior to resubmission.
Author Response
Dear Reviewer,
We are grateful for your time and expertise in reviewing our manuscript. We have carefully considered all of your suggestions and have made appropriate revisions accordingly. Your constructive feedback has undoubtedly enhanced the quality of our article.
Please see the attachment.
